# In Vitro Antioxidant and Fibroblast Migration Activities of Fractions Eluded from Dichloromethane Leaf Extract of *Marantodes pumilum*

**DOI:** 10.3390/life13061409

**Published:** 2023-06-17

**Authors:** Abbirami Balachandran, Stepfanie N. Siyumbwa, Gabriele R. A. Froemming, Morak-Młodawska Beata, Jeleń Małgorzata, Charlie A. Lavilla, Merell P. Billacura, Patrick N. Okechukwu

**Affiliations:** 1Department of Biotechnology, Faculty of Applied Sciences, UCSI University, Cheras, Kuala Lumpur 56000, Selangor, Malaysia; 1001233455@ucsiuniversity.edu.my; 2Department of Pathology and Microbiology, School of Medicine, Lusaka P.O. Box 50110, Zambia; ssiyumbwa@unmc.edu; 3Basic Medical Sciences, Faculty of Medicine and Health Sciences, Universiti Malaysia Sarawak (UNIMAS), Kota Samarahan 94300, Sarawak, Malaysia; rafgabriele@unimas.my; 4Faculty of Pharmaceutical Sciences, Department of Organic Chemistry, Medical University of Sílesia, Jagiellonska, Str. 4, 41-200 Sosnowiec, Poland; bmlodawska@sum.edu.pl; 5Chemistry Department, College of Science & Mathematics, Mindanao State University-Iligan Institute of Technology, Iligan City 9200, Lanao del Norte, Philippines; charliejr.lavilla@g.msuiit.edu.ph; 6Department of Chemistry, College of Natural Sciences & Mathematics, Mindanao State University-Main Campus, Marawi City 9700, Lanao del Sur, Philippines; merell.billacura@msumain.edu.ph

**Keywords:** diabetes, insulin resistance, natural products, reactive oxygen species, wound healing

## Abstract

(1) The complexity of diabetes and diabetic wound healing remains a therapeutic challenge because proper and systematic wound care and management are essential to prevent chronic microbial infection and mechanical damage to the skin. *Marantodes pumilum*, locally known as ‘Kacip Fatimah’, is an herb that has been previously reported to possess anti-inflammatory, analgesic, antinociceptive and antipyretic properties. The current study aims to assess the antioxidant and fibroblast cell migration activities of the fractions eluded from the dichloromethane extract of *M. pumilum* leaves. (2) The total antioxidant capacity of *M. pumilum* was assessed using the total proanthocyanidins and phosphomolybdenum assays, while DPPH, nitric oxide, hydrogen peroxide and superoxide free radical scavenging assays were tested to determine the antioxidant potential of *M. pumilum*. An in vitro scratch wound assay was performed to measure the fibroblast cell migration rate using normal and insulin-resistant human dermal fibroblast cells. (3) All *M. pumilum* fractions exhibited good antioxidant and fibroblast cell migration activity, among which fractions A and E displayed the greatest effect. (4) *M. pumilum*’s fibroblast migration activity could be attributed to its strong antioxidant properties along with its previously reported properties.

## 1. Introduction

Wound healing is the biological process by which a damaged or disrupted tissue is mended so that the tissue’s function can be restored, and future damage can be avoided. Diabetes mellitus (DM) is a disease that affects over 400 million individuals worldwide. Hyperglycaemia is a prominent symptom of diabetes, and it can lead to serious consequences. Impaired wound healing is one of the consequences of DM, and it affects roughly 20% of DM patients [1]. Hyperglycaemia is linked to wound-healing suppression due to reduced fibroblast cell migration and defective angiogenesis [2]. The physiological response to wound healing includes bleeding, vascular contraction with coagulation, complement activation and inflammatory response [3].

Wound healing is a three-step process that includes (1) inflammation, proliferation and remodelling before moving on to a more complex and well-organized interaction between diverse tissues and cells by overlapping these stages [4]. Any changes to any of these stages could slow the healing of the cutaneous wound. Traditional inflammation symptoms, such as pain, reddening and oedema, are frequently seen in wounds. Vasoconstriction and platelet aggregation begin shortly after injury, before the infiltration of leukocytes and T lymphocytes (inflammation stage). The main goal of the inflammatory stage is for neutrophils and macrophages to remove debris, injured tissue and germs. By producing proteolytic enzymes and reactive oxygen species (ROS), debris, damaged tissues and bacteria contribute to the antimicrobial defence system and the debridement of devitalized tissue [5]. The dermal fibroblast is a critical executor during wound healing, and its migration and proliferation are essential and rate-limiting steps to repair wounds due to its central role in the formation of granulation tissue. If granulation tissue formation is dysfunctional, wound healing may be delayed, or wounds may not heal at all [6].

Because ROS act as supplementary messenger signalling molecules and play a role in the defensive mechanism against invading germs, they are critical to wound healing [7]. An increase in neutrophils and reactive oxygen species (ROS) flood the antiprotease chemicals that normally protect tissue cells and the extracellular matrix (ECM), resulting in unfavourable outcomes, such as apoptosis and ECM breakdown [8]. Acute tissue damage caused by an increase in ROS levels may result in neoplastic transformation, which may impede the healing process due to damage to cellular membranes, DNA, proteins and lipids [9]. ROS at high levels can destroy fibroblasts and make skin lipids less flexible.

Antioxidants aid wound healing by neutralizing excess proteases and ROS produced by neutrophil build-up at the damaged site, as well as protecting protease inhibitors from oxidative damage [10]. The antioxidant system is made up of two types of antioxidants: (1) enzymatic antioxidants, such as superoxide dismutase, glutathione peroxidase, catalase and thioredoxin, that protect cells by catalysing ROS directly or indirectly, and (2) non-enzymatic antioxidants that protect cells by promoting antioxidative enzymes or directly processing the oxidative chain reaction [11]. Antioxidant capabilities are abundant in many plants that have traditionally been used to treat wounds [12].

*Marantodes pumilum* (formerly known as *Labisia pumila* var. alata), also known as “Kacip Fatimah,” is a blooming plant in the Myrsinaceae family that is popular among Malaysia’s Malay female community. It is used as a pain reliever, postpartum medication and to treat inflammatory conditions, such as rheumatism and fever, in folkloric medicine. *M. pumilum’s* crude extract and bioactive ingredients have been shown to exhibit anti-inflammatory, analgesic, antinociceptive, antioxidant and antipyretic properties in earlier research [13,14,15]. Bioactive components extracted from *M. pumilum*, such as phenolic acids and flavonoids, have been shown to exhibit antibacterial, antioxidant and wound-healing properties [16,17]. Because antioxidants are important for fibroblast cell migration in wound healing, this study aimed to assess the in vitro antioxidant and fibroblast cell migration properties of *M. pumilum* fractions of the dichloromethane (DCM) extract on normal and insulin-resistant human dermal fibroblast (HDF) cell line.

## 2. Materials and Methods

### 2.1. Biological Material

Dr Shamsul Khamis (coordinator of the biodiversity unit from the Laboratory of Natural Products (NATPRO), Institute of Bioscience in University Putra Malaysia) identified and collected *M. pumilum* leaves (68 g) from Sungai Perak Forest, UPM.

### 2.2. Preparation of Plant Material

The leaves were cleaned with distilled water, oven dried for 7 days at 36 °C and ground for 5 min with a mechanical grinder. The plant powder was macerated in DCM for two days with agitation at 150 rpm, then filtered and concentrated under vacuum in a Buchi-rotary evaporator at 40 °C.

### 2.3. Column Chromatography Purification

For partial purification of *M. pumilum* crude extract, a 42 × 2.5 cm vertical column was washed and constructed. To make the silica gel and extract mixture, 8 g of crude leaf extract was combined with 50 mL analytical-grade DCM and filtered three times using Whatman paper before being placed into a 4.8 g silica gel. The silica gel was air-dried overnight, yielding a final weight of 12.65 g of silica gel mix. The crude extract of *M. pumilum* was separated using hexane-ethyl acetate (Hx-EtOAc) in increasing percentages of ethyl acetate. The recovered eluents were concentrated using a rotary evaporator and kept at −20 °C until further investigation.

### 2.4. Antioxidant Studies

#### 2.4.1. Total Antioxidant Capacity (TAC)

The phosphomolybdenum assay was used to measure the total antioxidant capacity of the partly purified *M. pumilum* extracts [18]. Sulfuric acid (0.6 M in cold methanol), sodium phosphate and ammonium molybdate tetrahydrate were combined with 1 mg/mL of the sample. At 695 nm, the absorbance of the combination was measured. As a control, ascorbic acid was employed.

#### 2.4.2. Total Proanthocyanidins (TPA)

The total proanthocyanidin content was determined using the vanillin-HCl assay [19]. In a test tube, 1 mL of the sample (1 mg/mL) was added to 2.5 mL of 1% vanillin solution. For the control, 2.5 mL of methanol was used instead of the vanillin solution. Subsequently, 2.5 mL of 9 mol/L of hydrochloric acid (HCl) in methanol was added. The mixture was left to incubate for 20 min at 30 °C before reading the absorbance at 500 nm.

### 2.5. Free Radical Scavenging Studies

#### 2.5.1. 2,2-Diphenylpicrylhydrazyl (DPPH) Assay

The activity of the 2,2-diphenylpicrylhydrazyl (DPPH) radical was measured using the methods of previous studies [20,21]. In a test tube, 1 mL of the sample (50–500 µg/mL) and 1 mL of DPPH solution (7.8 mg in 100 mL methanol) were added together. The tubes were wrapped with aluminium foil and left to incubate in the dark for 30 min. The absorbance readings were measured at 517 nm.

#### 2.5.2. Nitric Oxide (NO) Assay

Nitric oxide (NO) activity was measured using a combination of sodium nitroprusside in phosphate buffer pH 7.4 and naphthylethylene diamine dihydrochloride (NEDD) [22]. The reaction mixture contained 1 mL of 10 mM sodium nitroprusside, 0.25 mL phosphate buffer saline and 0.25 mL of the sample (50–500 µg/mL). The mixture was incubated for 2 h at 25 °C. After incubation, 0.5 mL of the mixture was transferred into a new tube to which 1 mL of sulfanilic acid reagent (0.33% in 20% glacial acetic acid) was added. The tube was mixed thoroughly and left for 5 min. Lastly, 1 mL of 0.1% NEDD was added and left to stand for 30 min. The absorbance of a pink chromophore was observed at 540 nm against corresponding blank solutions.

#### 2.5.3. Hydroxyl Radical Inhibition Assay

The hydroxyl (OH^−^) radical scavenging activity was measured using the Haber–Weiss method [23]. To a 1 mL of reaction mixture, samples (50–500 µg/mL), 10 mM ferric chloride (FeCl_3_), 1 mM ethylenediaminetetraacetic acid (EDTA), 10 mM hydrogen peroxide (H_2_O_2_), 10 mM deoxyribose and 1 mM ascorbic acid were added. This mixture was incubated at 37 °C for 1 h and subsequently heated at 80 °C with the addition of 1 mL 0.5% thiobarbituric acid in 0.025 M sodium hydroxide and 0.02% butylated hydroxyanisole (BHA), as well as 1 mL of 10% trichloroacetic acid (TCA) in a boiling water bath for 45 min. The mixture was then allowed to cool before measuring the absorbance at 532 nm.

#### 2.5.4. Superoxide Radical Inhibition Assay

The superoxide (SO^−^) radical scavenging activity was determined by adding 50 mM phosphate buffer (pH 7.6), 20 µg riboflavin, 12 mM EDTA and 0.033 mg/mL nitro blue tetrazolium solution sequentially into a test tube. The reaction commenced with the addition of the samples (50–500 µg/mL) for 90 s before measuring the absorbance immediately at 590 nm. Ascorbic acid was used as the positive control [24].

### 2.6. Fibroblast Cell Migration Studies

#### 2.6.1. HDF Cell Culture and Differentiation

DMEM growth media (HIMEDIA, Mumbai, India) were used to grow normal HDF cells (ATCC PCS-201-012), which were incubated at 37 °C and 5% CO_2_ standard conditions. The culture media were changed to DMEM with 2% *v/v* horse serum (Gibco, Lower Hutt, New Zealand) and 1% antibiotic solution (penicillin-streptomycin) to stimulate differentiation. The cells were incubated for 7 days under standard conditions.

#### 2.6.2. Induction of Insulin Resistance

The induction method of insulin resistance in the fibroblast cells was slightly modified from the methods published by previous studies [25,26]. For 24 h, differentiated fibroblasts were given a medium containing 25 mmol/L glucose and 100 nmol/L insulin. After the 24 h treatment, the spent media were discarded, and the cells were washed in phosphate buffer saline (PBS) to remove debris before being treated with a new serum-free medium containing 5 mmol/L glucose only for 5 h. The spent media were discarded after 5 h and the cells were stimulated with 100 nmol/L insulin only in a new medium for another 30 min before treating the cells with respective controls and treatments.

#### 2.6.3. Scratch Wound Assay

Next, 70 µL of 3 × 10^5^ cells/mL cell suspension was planted into each well of the Ibidi culture insert in µ-Dish. For adhesion, the cells were cultured for at least 24 h under normal conditions. The next day, the insert was removed to check for attachment. The solution was removed after washing the cell layer with PBS to eliminate any non-attached cells. In the µ-Dish, 2 mL of each fraction was added. The µ-Dish was then placed on the microscope and rotated around until the gap was visible and imaged. Images were taken at 0, 4, 8 and 24 h during the observation.

### 2.7. Identification of Chemical Constituents in Fractions A and E

Among the five fractions, Fractions A and E were further studied to identify the bioactive compounds present in these fractions due to the highest antioxidant and fibroblast migrating activities. Fraction A was subjected to gas chromatography–mass spectrometry (GC-MS), whereas Fraction E was subjected to high-performance thin-layer chromatography (HPTLC) and liquid chromatography–mass spectrometry coupled with multiple reaction monitoring (LC-MS/MRM).

#### 2.7.1. GC-MS of Fraction A

The sample was dissolved in methanol (1 mg/mL) where 2 µL of the sample was injected into a GC-MS system (Agilent 5973 GCMS). An HP-5MS column (30 m × 0.25 µm) was used for the study. The injector temperature was 250 °C with the temperature program as follows: initial temperature 70 °C, increased by 10 °C/min to 300 °C for 6 min. The run time was 29 min with a 1 min post-run. The total flow rate was 54 mL/min with a septum purge flow of 3 mL/min.

#### 2.7.2. HPTLC of Fraction E

Twenty microlitres (20 µL) of the extracts was separately applied (samples and standard) onto the TLC plate with a 6 mm wide band spotted with an automatic TLC applicator Linomat V with N2 flow (Camag, Arlesheim, Switzerland), 8 mm from the bottom with instruction input defined from win-CATS-V 1.2.3 software. After sample application, the plates were developed in a 10 × 20 cm horizontal Camag twin glass chamber pre-saturated with the mobile phase (10 mL on each side) for 20 min at room temperature (25–27 °C). The mobile phase consisted of MeOH: EA (60:40 *v/v*). Linear ascending development was carried out until the 8 cm mark. The plates were observed after 30 min of air drying under the Camag UV visualizer (366 nm). Derivation was performed with β-Aminoethyl-diphenylborate for α and γ-pyrones (Neu’s reagent) spray, which was prepared by dissolving 1 g of diphenylborinic acid aminoethylester in 100 mL of 100% MeOH and was then blow dried. Video Scan software in fluorescence mode was used to quantify the plates post-derivation.

#### 2.7.3. LC-MS/MRM of Fraction E

To further confirm the bioactive compound present in Fraction E, the sample was analysed using this technique. A Hypersil Gold RP C18 column (2.1 mm I.D. × 100 mm; 3 µm) column was used with two solvents: 0.1% aqueous formic acid (FA) and acetonitrile (AC). The sample was prepared at 1 mg/mL using HPLC-grade methanol. The flow rate used in this experiment was 200 µL/min, and 10 µL injection volume was used. The MS was operated in the LTQ Ion Trap mass spectrometer (Thermo Scientific, Waltham, MA, USA) with U-HPLC system (Accela) MRM according to the manufacturer’s recommended operating conditions.

### 2.8. Statistical Analysis

The mean and standard deviation (*n* = 3) were used to present all the data. The EC_50_ values for the free radical scavenging investigations were calculated using linear regression analysis. The statistically significant differences between the *M. pumilum* and the controls were compared using a one-way analysis of variance (ANOVA) and post hoc testing.

## 3. Results

### 3.1. Column Chromatography Purification

The column chromatography separation produced five fractions, each with a different colour and yield as shown in Table 1. Fraction A was eluted from 100% hexane to a 90:10 ratio (Hx:EtOAc). Fraction B eluted between 80:20 (Hx:EtOAc) and 75:25 (Hx:EtOAc) (Hx:EtOAc). Fraction C eluted from 70:30 (Hx:EtOAc) to a 60:40 (Hx:EtOAc) ratio (Hx:EtOAc). From 60:40 (Hx:EtOAc) to 100% ethyl acetate, fractions D and E eluted. Using a rotary evaporator, the eluent solvents were evaporated to obtain solvent-free fractions.

### 3.2. Antioxidant Studies

The total antioxidant capacity (TAC) and total proanthocyanidins (TPA) of the partly purified DCM fractions A-E from *M. pumilum* were determined in this investigation, as indicated in Table 2. Fraction E had the most proanthocyanidins and antioxidant capacity followed by Fraction D for TAC and Fraction A for TPA.

The *p* values for TAC and TPA were *p* = 0.001 and *p* = 0.004, respectively, in a one-way ANOVA study. This demonstrates that the population means differed greatly from one another. TPA’s post hoc (Tukey’s HSD) test revealed that Fraction E was considerably higher than all other eluted fractions, with a *p* value of 0.001. In comparison to the other fractions, Fraction E appears to have the most potential for scavenging free radicals. Fraction B, on the other hand, was considerably lower than fractions D (*p* = 0.037) and E (*p* = 0.003), whereas Fraction E was significantly higher than Fraction C (*p* = 0.026) in the TAC post hoc test.

The correlation of TAC against TPA was investigated and plotted, as shown in Figure 1. The correlation graph showed a positive correlation (R^2^ value = 0.8557) of TAC and TPA, indicating that the total antioxidant capacity in these fractions is linked to the total proanthocyanidins found in each fraction.

#### Free Radical Scavenging Studies

Table 3 shows the EC_50_ values of *M. pumilum* fractions derived from the DPPH scavenging assay. The EC_50_ values revealed that to scavenge DPPH radicals, Fractions B and C, with EC_50_ values of 266.2 ± 0.23 µg/mL and 72.2 ± 0.06 µg/mL, respectively, required greater concentrations. Fractions A, B, D and E exhibited potential in scavenging NO radicals when compared to the standard (ascorbic acid), with EC_50_ values of 264.1 ± 0.12 µg/mL, 406.9 ± 0.07 µg/mL, 195.6 ± 0.14 µg/mL, 109.9 ± 0.12 µg/mL and 328.9 ± 0.10 µg/mL, respectively.

The fractions’ EC_50_ values against OH^−^ radicals revealed that their scavenging activity was notably different from the standard. This means that the fractions’ concentrations should be higher than the amounts utilised in this experiment if they are to have strong OH^−^ radical scavenging activity. However, Fraction E showed the closest scavenging activity to the standard, with EC_50_ values of 193.3 ± 0.24 µg/mL and 54.98 ± 0.04 µg/mL, indicating that there are chemicals in this fraction that can scavenge OH^−^ radicals.

The SO^−^ radical scavenging assay revealed similar results to the OH^−^ scavenging assay, with all fractions yielding significantly different values from the standard, indicating that the doses utilised in this experiment were too low to exhibit strong SO^−^ radical scavenging activity. However, Fraction A had the closest EC_50_ value of 162.8 ± 10.7 µg/mL to the standard (EC_50_ value of 103.9 ± 0.94 µg/mL), implying that Fraction A included chemicals that are potentially significant scavengers of SO^−^ radicals at low concentrations.

### 3.3. Fibroblast Cell Migration Studies

The scratch wound assay showed that HDF cells treated with the fractions were able to stimulate the migration of fibroblasts and close the gap in a period of 24 h. Figure 2 shows a comparison of fibroblast migration in untreated cells against cells treated with different fractions of *M. pumilum* in normal HDF cells. It was observed that the untreated cells did not seal the gap, even after 24 h, whereas cells treated with *M. pumilum* fractions were able to seal the gap after 24 h.

The distance travelled by the fibroblast cells in different treatment groups was measured through the observed gap distance between the fibroblast cells of the scratch wound assay at the initial point (0 h), after 4 h, after 8 h and after 24 h. Figure 3a shows the gap distance of normal HDF cells in different treatment groups. Although all fractions were able to completely migrate and seal the gap, Fractions A and E had the fastest fibroblast migration rate compared to the negative control, with a percentage of 83.34% and 85% by the eighth hour, respectively, as shown. Fractions B, C and D were only able to migrate at half the rate of Fractions A and E by the eighth hour, with percentages of 38.93%, 40.22% and 50.74%, respectively. 

Comparatively, the fibroblast migration rate of all the fractions in the insulin-resistant HDF cells was much slower than that of the normal HDF cells (Figure 3b). However, among all the fractions, Fractions A and E completely sealed the gap by the 24th hour. The fibroblast migration rate of Fractions A and E was only 20.04% and 35.8% by the 8th hour but, interestingly, they achieved a 100% migration rate by the 24th hour, indicating that Fractions A and E have strong fibroblast migration activity in both normal and insulin-resistant HDF cells. On the other hand, Fractions B, C and D were not able to seal the gap, even by the 24th hour, with a total migration rate below 50%. This shows that Fractions B, C and D work well in normal conditions but are poor stimulants of fibroblast migration in an insulin-resistant state.

### 3.4. Identification of Major Compounds Present in Fractions A and E

Fractions A and E were further subjected to identify the major compounds present in each of these fractions, as the antioxidant and fibroblast migration activities of these fractions were the strongest among all the fractions. In Fraction A, 29 compounds were identified, as shown in Table 4, with their different retention times, as shown in Figure 4.

On the other hand, the HPTLC and LC-MS/MRM results revealed naringin as the major compound in Fraction E, while catechin was detected in trace amounts, as shown in Figure 5 and Figure 6.

## 4. Discussion

The proliferation of fibroblast cells is very important in wound healing, as the proliferative phase is characterized by the development of the granulation tissue to cover the wound area for the complete tissue repair and the angiogenesis process [5]. However, in chronic diseases such as diabetes, the proliferation and migration of fibroblasts are slowed down by many factors, including chronic oxidative stress, inflammation, insulin resistance, decrease in the expression of several growth factors and the overexpression of the matrix metalloproteinases (MMPs), such as MMP-2 and MMP-9 [27,28,29]. This condition is linked to a delayed wound-healing process, as seen in diabetic patients, which is a major clinical issue with a yearly economic impact in the billions of dollars [30]. Many reports have shown that antioxidants are associated with the stimulation of fibroblast migration. This process may be through alleviating oxidative stress, increasing expression of growth factors, inhibiting MMPs and stimulating internal antioxidant enzymes [13,14,15,16,17,31,32]. In this study, the authors hypothesise that the mechanism of action of fractions A to E may be through any of those mentioned above.

In the current research, Fractions A to E showed strong antioxidant and fibroblast migration activities, which is in consonance with other established reports. In addition, a positive correlation was observed between TAC and TPA, indicating that the antioxidant capacity was dependent on the total proanthocyanidins present in these fractions. Previous reports have linked antioxidants with fibroblast migration. For instance, Boakye et al. (2018) reported that antioxidants stimulate increased cell proliferation, barrier formation and extracellular matrix (ECM) formation on dermal fibroblasts and keratinocytes [33]. Another study showed that antioxidants inhibit overexpression of MMP-2 and MMP-9, which enhances the growth of fibroblasts and keratinocytes [34]. The increase in fibroblast growth by antioxidants may be through the expression of growth factors, such as fibroblast growth factor (FGF), vascular endothelial growth factor (VEGF), platelet-derived growth factor (PDGF) and excess protease activity [35,36,37]. Apart from that, the reductions in lipid peroxidation, hydrogen peroxide and nitric oxide activities play a role in the augmentation of fibroblasts as well [38]. Another mechanism of antioxidants in improving fibroblast proliferation is through reductions in ROS produced by the neutrophils and macrophages that accumulate during the inflammatory stage of the wound-healing process that could potentially kill fibroblast cells at high concentrations [32,39].

The current paper also highlights the different classes of phytochemicals (phenolics, flavonoids, hydrocarbons and polyphenols) that were identified in Fractions A and E. Most of these compounds have been reported to possess antioxidant and fibroblast migration effects, suggesting that the activities observed in these fractions may be due to these phytochemicals. For instance, polyphenols, flavonoids and tannins, such as naringin, resveratrol, ferulic acid, geraniin, syringic acid and chlorogenic acid, have been found to increase fibroblast proliferation and keratinocyte migration as well as enhance hydroxyproline and glutathione levels [33,38,40,41,42,43,44,45]. Several studies have also reported that antioxidants increase the expression of growth factors VEGF and FGF to boost fibroblast and keratinocyte proliferation [46,47]. Apart from this, activation of sirtuins by polyphenols increases the proliferation of fibroblasts by increasing the replicative lifespan (RLS) of fibroblasts. Sirtuins are a family of deacetylase enzymes that play a role in aging, transcription, apoptosis, inflammation and stress resistance [48,49]

Polyphenols have been shown to prolong the RLS of fibroblasts through the activation of sirtuins [50]. Resveratrol, a polyphenol found in grape skin and wines, was discovered to mimic calorie restriction by stimulating SIRT1, although this mechanism remains unclear [51]. Another study showed that 5 µM of resveratrol was found to prolong the RLS of fibroblasts [44]. Apart from this, flavonoids, such as quercetin and caffeic acid, have been shown to activate proteasomes and have a better effect in increasing the gene expression of SIRT1 and SIRT6 than resveratrol [52,53]. Another potent antioxidant, epigallocatechin gallate (EGCG), which is commonly found in green and black teas, has also been shown to increase the antioxidant activity of superoxide dismutase and catalase, which decreased ROS levels in replicatively advanced fibroblasts [54]. Moreover, anthocyanidins, such as cyanidin and malvidin, and ellagitannins such as urolithin A prolonged the RLS of fibroblasts. Urolithin A significantly increased type I collagen and reduced the expression of MMP-1 in old human dermal fibroblasts along with reducing intracellular ROS, which could be due to the activation of the Nrf-2 mediated antioxidant response [55,56,57].

Furthermore, various plant extracts, including *B. globosa, C. odorata, P. appendiculatum*, *B. ferruginea, C. infortunatum* and *M. esculenta,* have been proven to possess antioxidant effects, which, in turn, stimulate fibroblast and keratinocyte migrations, upregulate ECM protein and basement membrane component production by keratinocyte and inhibit collagen lattice contraction through fibroblasts [58,59,60,61,62,63,64,65,66]. Naturally occurring antioxidants such as alpha-lipoic acid (ALA; also known as thioctic acid) have been demonstrated to exhibit antioxidant activity by directly interacting with free radicals, counteracting lipid and DNA damage caused by oxygen exposure or by recycling vitamin E, thereby increasing the plasma’s overall antioxidant status. ALA has also been shown to have an inhibitory effect on pro-inflammatory cytokines [67]. It has been shown that adding ALA to hyperbaric oxygen (HBO) therapy improves wound healing by removing the negative side effects of oxygen exposure and so speeds up the healing process [68].

Interestingly, the different types of drug delivery systems influence the potential of the drug to stimulate the proliferation and migration of fibroblasts, resulting in efficient wound-healing progression. A study by Zhao et al. (2020) reported that the drug-loaded ROS-scavenging hydrogel promotes wound closure by decreasing the ROS level and upregulating M2 phenotype macrophages around the wound. Such hydrogels formed in wounds also increased the release of granulocyte macrophage colony-stimulating factor to fight against external bacteria and improve the wound closure [69]. Moreover, the use of core–shell nanofibers and segmented polyurethanes (SPUs) that are able to carry multiple drugs has shown promising efficiency in wound dressing and wound closure applications [70,71]. Similarly, Lee et al. (2022) studied the use of poly-(lactic-co-glycolic acid) (PLGA)-based saxagliptin membranes, which exhibited strong antioxidant activity, cellular granulation and functionality in diabetic wound closure [72]. These studies have given deeper insights into researching the suitable type of drug delivery system for Fractions A and E to increase the competency of the fractions in the treatment of wounds.

## 5. Conclusions

In summary, Fractions A and E of *M. pumilum* displayed strong antioxidant properties through the DPPH, NO, hydroxyl and superoxide radical scavenging assays, as well as greatly stimulating fibroblast migration. This stimulation could be due to the presence of different phytochemicals in these fractions that elicit strong antioxidant activities, which influence the migration activity of the fibroblasts. Although the exact mechanism of action of the antioxidants in stimulating fibroblast migration remains unclear, the current findings suggest that the mechanism may be through: (1) a reduction in oxidative stress caused by lipid peroxidation, nitric oxide and hydrogen peroxide accumulation; (2) stimulating the efficiency of endogenous antioxidant system; (3) increase in expression of FGF, VEGF and PDGF; and (4) inhibition of MMP-2 and MMP-9. The authors believe that the inhibition of nitric oxide, hydroxyl and superoxide radicals may have inhibited MMP-2 and MMP-9 and enhance hydroxyproline and glutathione levels. However, further studies are required to elucidate the mechanism of action of the antioxidants on the stimulation of fibroblast migration using different in vitro and in vivo models.

## Figures and Tables

**Figure 1 life-13-01409-f001:**
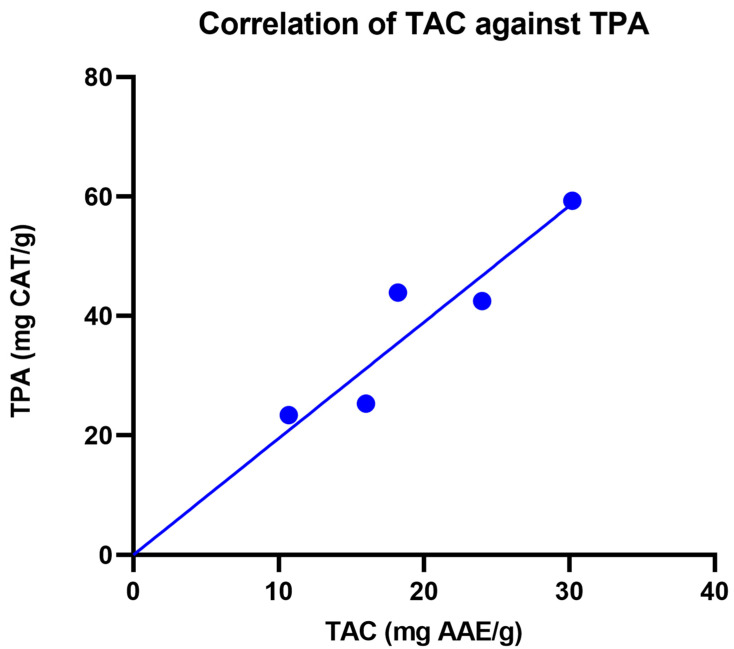
Correlation graph of TAC versus TPA of fractions A to E eluded from *M. pumilum* DCM leaf extract.

**Figure 2 life-13-01409-f002:**
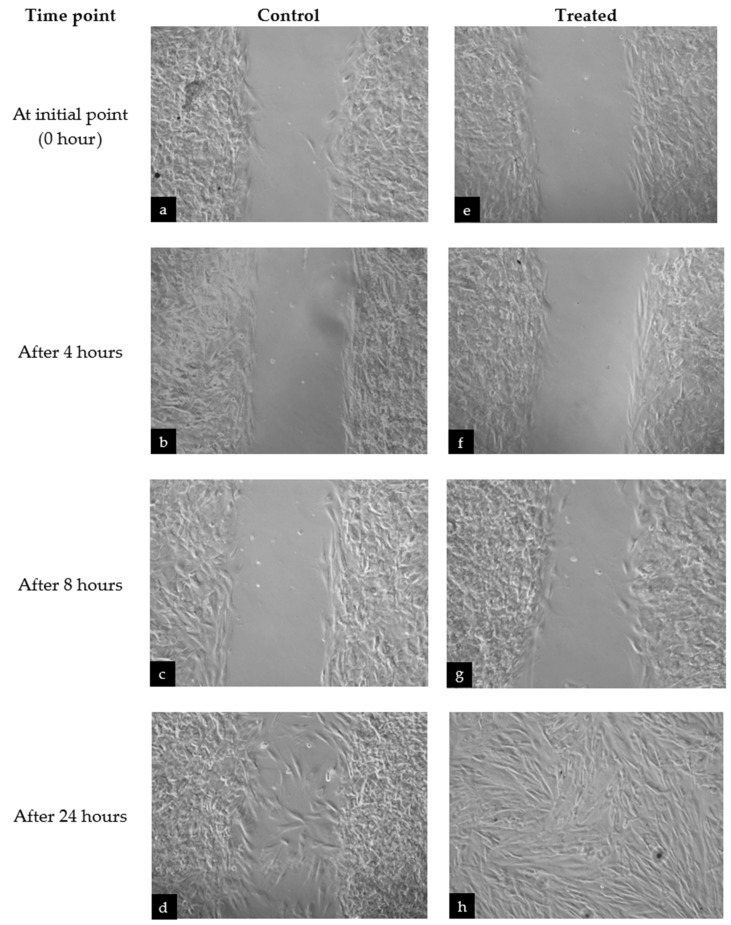
A sample of normal fibroblast cell migration images taken at different time points to compare between (**a**–**d**) the negative control and (**e**–**h**) cells treated with Fraction E of *M. pumilum*. All fractions displayed similar migration images and, thus, the pictures of Fraction-E-treated cells were selected to represent the fibroblast migration of all fraction-treated cells observed at different time points.

**Figure 3 life-13-01409-f003:**
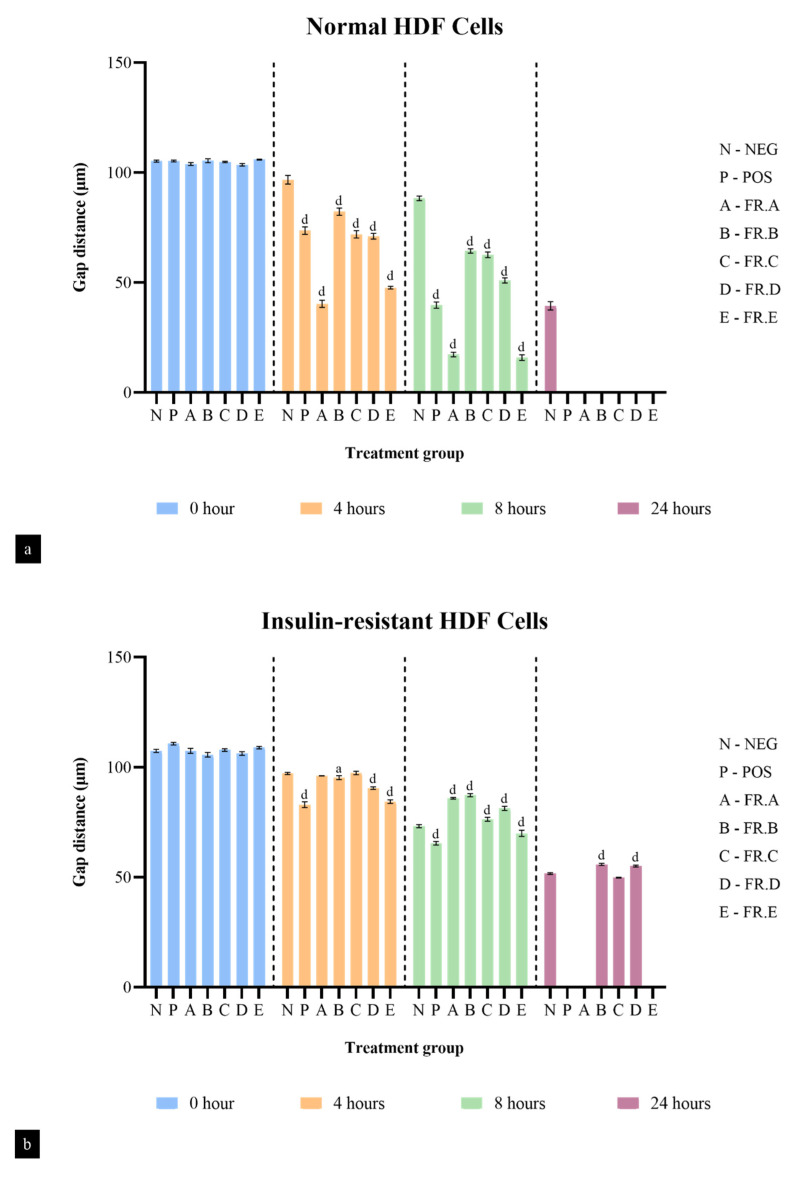
Gap distance of (**a**) normal HDF cells and (**b**) insulin-resistant HDF cells of different treatment groups from the in vitro scratch wound assay. One-way ANOVA statistical comparison is represented by letters ‘a’ as *p* < 0.05 and ‘d’ as *p* < 0.0001 against the negative group of the respective time points. NEG—negative control; POS—positive control; FR. A up to FR. E—fractions A to E.

**Figure 4 life-13-01409-f004:**
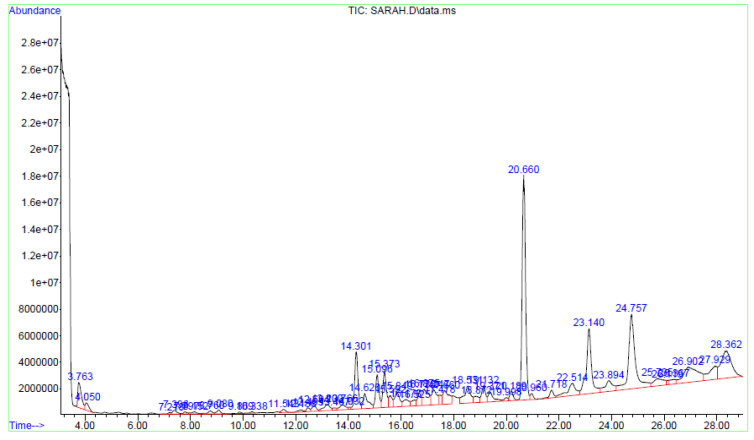
GC-MS analysis of Fraction A from the partially purified DCM leaf extract of *M. pumilum*.

**Figure 5 life-13-01409-f005:**
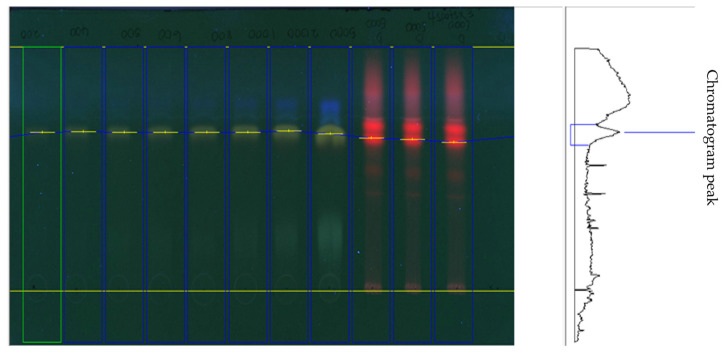
HPTLC analysis of Fraction E showed Naringin in the plant extract.

**Figure 6 life-13-01409-f006:**
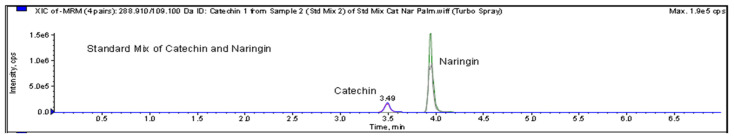
Targeted LC-MS/MRM analysis of Fraction E against naringin and catechin.

**Table 1 life-13-01409-t001:** Elution colour and yield of *M. pumilum* fractions.

Fractions	Colour of Eluent	Dry Weight (g)	Yield Percentage (%)
Fraction A	Yellow	0.878 ± 0.05	10.99 ± 0.03
Fraction B	Dark green	1.006 ± 0.02	12.60 ± 0.05
Fraction C	Green	0.548 ± 0.03	6.86 ± 0.08
Fraction D	Orange	0.760 ± 0.03	9.52 ± 0.04
Fraction E	Orange	0.554 ± 0.01	6.94 ± 0.01

**Table 2 life-13-01409-t002:** Total antioxidant capacity and total proanthocyanidins in each fraction.

Fractions	TAC (mg AAE/g)	TPA (mg CAT/g)
Fraction A	18.2 ± 5.29	43.9 ± 1.04 ^a^
Fraction B	10.7 ± 1.16 ^bc^	23.4 ± 3.41 ^a^
Fraction C	16.0 ± 6.93 ^b^	25.3 ± 1.00 ^a^
Fraction D	24.0 ± 1.00	42.5 ± 0.46 ^a^
Fraction E	30.2 ± 2.00	59.3 ± 1.06

All values are expressed as mean ± SD (*n* = 3). TAC was expressed in milligram ascorbic acid equivalent per gram (mg AAE/g) whereas TPA was expressed as milligram catechin equivalent per gram (mg CAT/g). One-way ANOVA analysis indicates that superscript letters ^a^ refers to *p* < 0.002, ^b^ refers to *p* < 0.05 and ^c^ refers to *p* < 0.005.

**Table 3 life-13-01409-t003:** The radical scavenging activity of *M. pumilum* fractions.

Assays	EC_50_ (µg/mL)					
	Standard	Fraction A	Fraction B	Fraction C	Fraction D	Fraction E
DPPH	0.05 ± 0.02	27.21 ± 4.48	266.53 ± 51.16 ^c^	72.50 ± 11.08 ^d^	4.63 ± 3.63	2.88 ± 0.95
NO	328.9 ± 24.72	464.0 ± 5.27 ^d^	406.9 ± 5.48 ^d^	314.4 ± 6.69	195.7 ± 12.34 ^d^	109.8 ± 3.69 ^d^
OH¯	54.98 ± 5.47	239.3 ± 6.87 ^d^	398.5 ± 9.6 ^d^	289.3 ± 7.91 ^d^	198.6 ± 8.07 ^c^	193.3 ± 3.88 ^b^
SO¯	103.9 ± 0.94	162.7 ± 10.7 ^a^	703.0 ± 10.9 ^d^	561.7 ± 53.6 ^d^	260.3 ± 15.2 ^d^	413.1 ± 7.84 ^d^

All values are expressed as mean ± SD (*n* = 3). The superscript letters represent the one-way ANOVA statistics comparison where ^a^ refers to *p* < 0.05, ^b^ refers to *p* < 0.01, ^c^ refers to *p* < 0.001 and ^d^ refers to *p* < 0.0001.

**Table 4 life-13-01409-t004:** List of compounds identified in Fraction A.

No	Compounds	Molecular Weight (MW)	Percentage Amount (PA)	Retention Time (mins)
1.	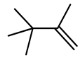	2,3,3-trimethyl 1-butene	98.186	2.16	3.762
	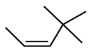	4,4-dimethyl-(Z)-2-pentene			
2.	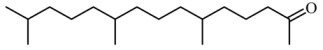	6,10,14-trimethyl-2-pentadecanone	268.478	4.46	14.302
3.	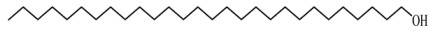	1-octacosanol	506.768	2.61	15.097
	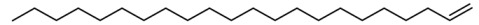	1-docosene	308.585		
	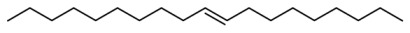	9-nonadecene	266.505		
4.	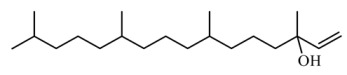	Isophytol	296.351	2.96	15.372
5.	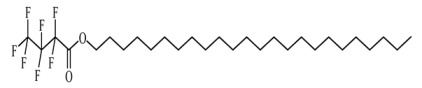	Tetracosyl heptafluorobutyrate	550.676	2.38	16.877
6.	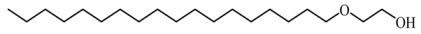	Ethanol, 2-(octadecyloxy)-	314.546	2.03	17.661
	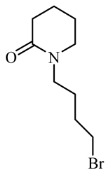	1-(4-bromobutyl)-2-piperidinone	234.133		
	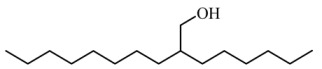	1-decanol, 2-hexyl-	242.441		
7.	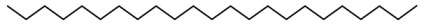	Tricosane	324.627	2.56	18.530
8.	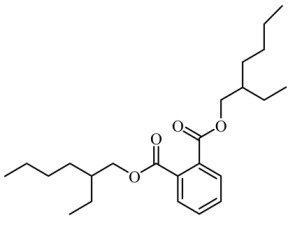	Bis(2-ethylhexyl) phthalate	390.556	17.80	20.659
	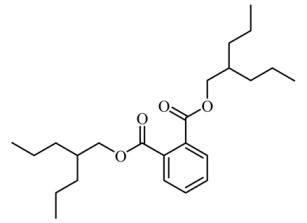	Bis(2-propylpentyl) phthalate			
9.	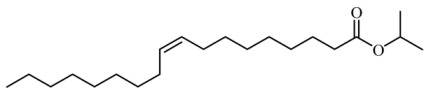	i-propyl 9-octadecenoate	324.541	2.46	22.513
	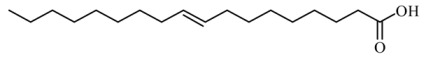	9-octadecenoic acid	282.461		
10.	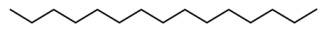	Pentadecane	212.415	7.89	23.142
	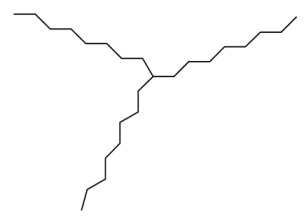	Heptadecane, 9-octyl-	352.680		
	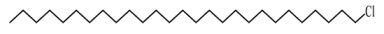	Heptacosane, 1-chloro-	415.179		
11.	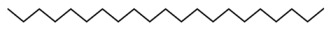	Heneicosane	296.574	11.72	24.756
	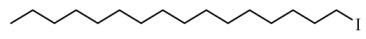	Hexadecane, 1-iodo-	352.338		
	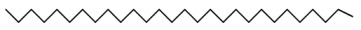	Octacosane	394.760		
12.	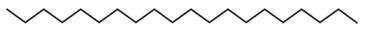	Eicosane	282.548	5.25	26.902
13.	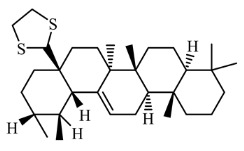	1,3-dithiolane-2-(28-norurs-12-en-17-yl)-	500.885	2.65	27.932
	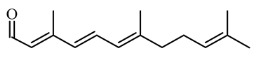	3,7,11-trimethyl-dodeca-2,4,6,10-tetraenal	218.335		
	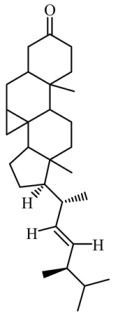	(22E)-3’,7β-dihydrocycloprop[7,8]-5α-ergost-22-en-3-one	NR		
14.	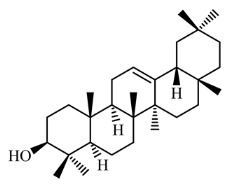	β-amyrin	426.717	6.84	28.361
	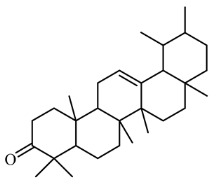	Urs-12-en-3-one	424.702		

## Data Availability

Data are contained within the article.

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
