# Peer review of "In Vitro Antioxidant and Fibroblast Migration Activities of Fractions Eluded from Dichloromethane Leaf Extract of Marantodes pumilum"

_life, 2023, doi:10.3390/life13061409_

Round 1

Reviewer 1 Report

Journal: Life

Title: In vitro Antioxidant and Fibroblast Migration Activities of Fractions Eluded from Dichloromethane Leaf Extract of Marantodes pumilum

        The article discusses a study on Marantodes pumilum's potential wound-healing properties. It assessed the antioxidant and fibroblast cell migration activities of fractions from the plant's leaves using various assays. All fractions showed good antioxidant and fibroblast cell migration activities, with fractions A and E displaying the greatest effect.

         By reading the overall manuscript, I suggest revision the following queries, 

Authors may wish to consider citing the following reference, as it could potentially enhance the strength of the present manuscript, particularly in regards to the comparison of various manufacturing methods.

(1) Sci. Rep. 2019, 9, 12640.

https://doi.org/10.1038/s41598-019-49132-x

(2) Nanomaterials 2022, 12(21), 3740.

https://doi.org/10.3390/nano12213740

(3)Polymers 2020, 12(12), 2882.

https://doi.org/10.3390/polym12122882

Author Response

See as attached 

Reviewer 2 Report

My comments and suggestions are represented in Word file

Author Response

See as attached 

Reviewer 3 Report

The current study assesses the antioxidant activity and wound healing potential of five fractions eluded from the dichloromethane (DCM) leaf extract of Marantodes pumilum. The author focused on antioxidant activity of the fraction. They performed several antioxidant chemical assays and found major bioactive compounds present in each fraction using GCMS and LCMS. However, there are some critical issues to be examined.

First, the wound healing model used in this study was Ibidi culture insert. This equipment left a space between cells, but not caused oxidative damage to the cells. Therefore, the activity of fractions only can imply the fibroblast migration induction. It’s hard to link the antioxidant activities of fraction with the fibroblast migration induction. The description of fraction with antioxidant activities benefited the wound healing should be modified. Therefore, the author should use another wound healing model with oxidative damage in vitro or in vivo, to examine the benefit of antioxidant activities of fractions.

Second, all the cell migration images should be included in the supplementary data.

Author Response

See as attached 

Round 2

Reviewer 2 Report

All comments and suggestions are in attached file (in red).

Author Response

See as attached 

Reviewer 3 Report

1.     The author made some explanation of fibroblast migration induction by oxidative activities in the discussion. That provided some evidences to support their findings. That will convince the audience.

2.     The images of cell migration treated with each fraction (A – E) are highly suggested to be included, at least in the supplementary data.

Author Response

as attached 
